# Pulmonary Nodule in a Patient with Oral and Lung Cancer: Cryptococcus Infection

**DOI:** 10.3390/dj7040102

**Published:** 2019-10-23

**Authors:** Kenji Yamagata, Chikako Hirano, Naomi Kanno, Fumihiko Uchida, Satoshi Fukuzawa, Toru Yanagawa, Hiroki Bukawa

**Affiliations:** Department of Oral and Maxillofacial Surgery, Institute of Clinical Medicine, Faculty of Medicine, University of Tsukuba, 1-1-1 Tennodai, Tsukuba, Ibaraki 305-8575, Japan; clover_cherry_1224@yahoo.co.jp (C.H.); greened_amethyst829@hotmail.com (N.K.); uchiyamada1031@yahoo.co.jp (F.U.); fukuzawa.satoshi@twmu.ac.jp (S.F.); ytony@md.tsukuba.ac.jp (T.Y.); bukawah-cuh@umin.ac.jp (H.B.)

**Keywords:** pulmonary nodule, oral cancer, lung cancer, Cryptococcus infection, antifungal agent

## Abstract

Pulmonary nodules are frequently considered to be a metastatic disease or primary lung tumors in oral cancer patients. We present a case of pulmonary cryptococcosis in a 68-year-old man with oral and lung cancer. This lung cancer was treated with thoracoscopic resection of the right inferior lobe and mediastinal lymph node dissection. Lower gingival cancer was treated with a mandibulectomy, neck dissection, and reconstruction after chemoradiotherapy. A 20 mm cavitary nodule appeared at the left lung S6 one-month after surgery, during post-operative computed tomography. Thoracoscopic partial resection of the left inferior lobe was performed under the suspicion of lung metastasis. Pathology results revealed a pseudo-epithelial granuloma with necrosis and many yeast-shaped fungi with capsules. A pathological diagnosis of Cryptococcus infection was made. The patient was prescribed the antifungal agent fosfluconazole, which was administered intravenously for 1 week and intraoral fluconazole for 12 months. No recurrence of the Cryptococcus infection has been noted after 1.5 years.

## 1. Introduction

Pulmonary nodules are frequently considered to be the primary lung tumors or metastatic disease in oral cancer patients. A primary lung tumor is the most common diagnosis for patients with solitary pulmonary nodules. On the other hand, multiple nodules are suspected metastatic lesions rather than primary lung tumors. The rate of distant metastasis from oral squamous cell carcinoma (OSCC) is reported to be about 10%, and the lung was the most common site of metastasis [1,2]. Distant metastasis from lung cancer was reported at 46.4%. The frequent metastatic sites were the nervous system, bone, liver, respiratory system, and adrenal gland; the respiratory system was the site of metastasis in 8.6% of all patients [3]. Cryptococcus neoformans is an encapsulated saprophytic fungus found worldwide in soils contaminated by bird excreta. Pulmonary cryptococcosis is a rare infection that can be lethal in immunocompromised patients and behaves as an opportunistic invasive fungal infection [4]. Therefore, it is important for clinicians to differentiate between malignant pulmonary nodules and pulmonary cryptococcosis [5].

There has been only one report on a patient with head and neck cancer [6]. We present here a rare case of pulmonary cryptococcosis in a patient with double cancer in the lower gingiva and the lung, in which the pathological diagnosis and treatment resulted in a good outcome.

## 2. Case Report

A 68-year-old Japanese man referred to the Department of Oral and Maxillofacial Surgery at the University Hospital of Tsukuba, complaining of a mass in the lower gingiva, one month after extraction of his frontal teeth. His medical history revealed diabetes mellitus, hypertension, hydrocephalus, and cerebral bleeding. He had no history of animal breeding. His general condition was good, and his face was symmetrical without trismus. The regional lymph nodes were not swollen. Intra-oral examination showed an irregular surface and an elastic hard mass with a necrotic ulcer between the right second premolar and the left first premolar, extending to the right floor of the mouth and measuring approximately 29 × 26 mm (Figure 1).

Short TI inversion recovery (STIR) sequence magnetic resonance imaging (MRI) showed a 34 × 31 × 21 mm heterogeneous, high-signal mass in the lower gingiva towards the floor of the mouth. Positron-emission tomography (18F-fluorodeoxy-glucose) combined with computed tomography (18F-FDG PET/CT) revealed the FDG uptake in the mass located at the lower gingiva, towards the floor of the mouth, with a standard uptake volume (SUV) max of 19.6. Chest X-rays revealed a 40mm, high-density mass nodule at the lower lobe of the lung. The CT depicted a high-density mass in the right S10, measuring 60 × 35 mm (Figure 2). The clinical diagnosis was lower gingival cancer (T4aN0M0, Stage IV) and lung cancer (T3N0M0, Stage IIA). At first, lung cancer was treated with thoracoscopic resection of the right inferior lobe and mediastinal lymph node dissection. The pathological diagnosis was squamous cell carcinoma (SCC), and the post-surgical course was uneventful. Chemoradiotherapy with a radiotherapeutic dosage of 41.4 Gy and administration of cetuximab for the gingival carcinoma started one month after lung surgery. After that, a supraomohyoid neck dissection, mandibulectomy, and reconstruction with a rectus abdominis musculocutaneous flap were performed under general anesthesia. The pathological diagnosis was SCC, and the post-surgical course was uneventful.

After one month, a 20 mm cavitary nodule appeared in the left lung S6 during the post-operative CT, and a thoracoscopic partial resection of the left inferior lobe was performed while suspecting lung metastasis (Figure 3). The resected specimen was a white solid mass with a charcoal powder deposition, measuring 20 × 15 × 11 mm (Figure 4). The pathology results revealed a pseudo-epithelial granuloma with necrosis and many yeast-shaped fungi with capsules. Encapsulated forms of Cryptococcus were revealed by Groccot staining, and a pathological diagnosis of Cryptococcus infection was made (Figure 5).

The cerebrospinal fluid was not examined because of the absence of symptoms in the central nervous system (CNS). An antifungal agent, fosfluconazole, was administered intravenously to the patient for 1 week, and intraoral fluconazole was administered for 12 months. The Cryptococcus infection did not recur after 1.5 years.

## 3. Discussion

The most common symptoms of cryptococcus infections are cough, fever, and dyspnea. On the other hand, approximately one-third of immunocompromised patients are asymptomatic [7]. Pulmonary nodules are the most common visual indicator of cryptococcosis, but these nodules are not specific to pulmonary cryptococcosis. They can mimic lung cancer, pulmonary tuberculosis, bacterial pneumonia, and other pulmonary mycoses, both clinically and radiologically [5]. The patient should be treated promptly after the diagnosis of pulmonary cryptococcosis with an antifungal therapy. This antifungal therapy responds well in most patients. Our patient was prescribed with the antifungal agent fosfluconazole, which was administered intravenously for 1 week and with intraoral fluconazole for 12 months, and no recurrence of Cryptococcus infection was observed thereafter.

Cryptococcus infection is an opportunistic infection that predominantly affects patients whose immune systems are compromised by HIV infection, solid organ transplantation, or under the clinical use of potent immunosuppressive agents, such as cancer chemotherapy, monoclonal antibodies, and corticosteroids [7]. Although leukopenia did not occur as a consequence of cetuximab treatment, our patient was immunocompromised because of consecutive lung surgeries, chemoradiotherapy, and oral reconstructive surgery. 

Cryptococcosis in cancer patients is a rare condition and appears to have a relatively good diagnostic and therapeutic outcome. In 19 cancer patients with pulmonary cryptococcosis, lung malignancy was the most frequent initial impression. These patients were not correctly diagnosed before performing the serum culture or tissue biopsy [8]. Image diagnosis and histopathologic diagnosis are useful for correct diagnosis, to ensure that an appropriate treatment is administered. In our case, we suspected lung metastasis from lung cancer because of N0 oral cancer, so a thoracoscopic partial resection of the left inferior lobe was performed. Although there has been only one report of a patient with head and neck cancer [6], pulmonary nodules in oral cancer patients may not be lung metastases or lung cancer, as occurred in our patient. 

The incidence of Cryptococcus infections in the CNS is reported to be spectacularly higher in immunocompetent patients. A lumbar puncture is necessary for pulmonary cryptococcosis patients with a clinically poor condition. Worse clinical patients show neurologic signs of disease progression or more than 1:250 high serum antigen titers [9]. Cerebrospinal fluid was not examined in our patient because of the absence of symptoms related to CNS infection. 

In conclusion, we presented a case of oral and lung cancer with a pulmonary nodule, which was resected as metastatic and diagnosed as a cryptococcus infection. The diagnosis of pulmonary Cryptococcus infection from various lung diseases is significant for accurate management with mycological examination of the lung tissue.

## Figures and Tables

**Figure 1 dentistry-07-00102-f001:**
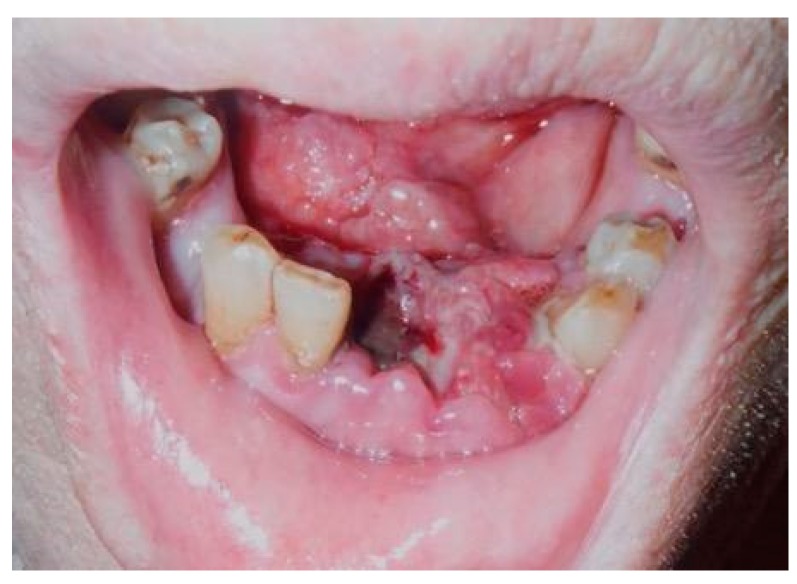
Necrotic ulcer. Intra-oral examination shows an irregular surface and elastic hard mass with a necrotic ulcer, which measures approximately 29 × 26 mm.

**Figure 2 dentistry-07-00102-f002:**
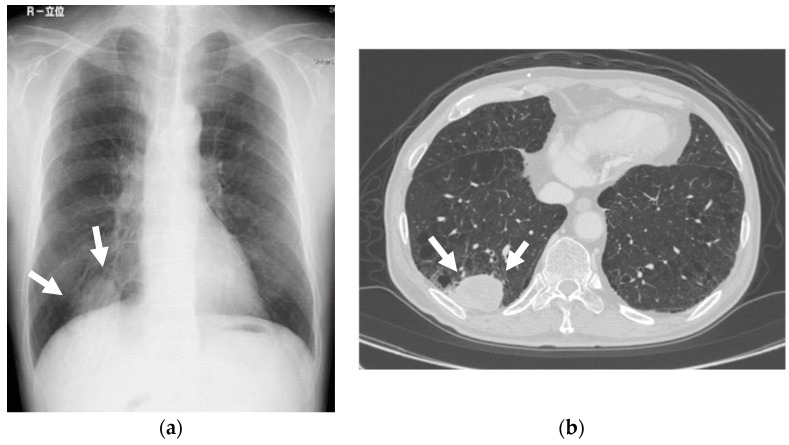
Mass nodule. Chest X-rays revealed a 40 mm high-density mass nodule at the right lower lobe of the lung. The CT depicted a high-density mass in the right S10, measuring 60 × 35 mm. (**a**) Chest X-rays; (**b**) CT.

**Figure 3 dentistry-07-00102-f003:**
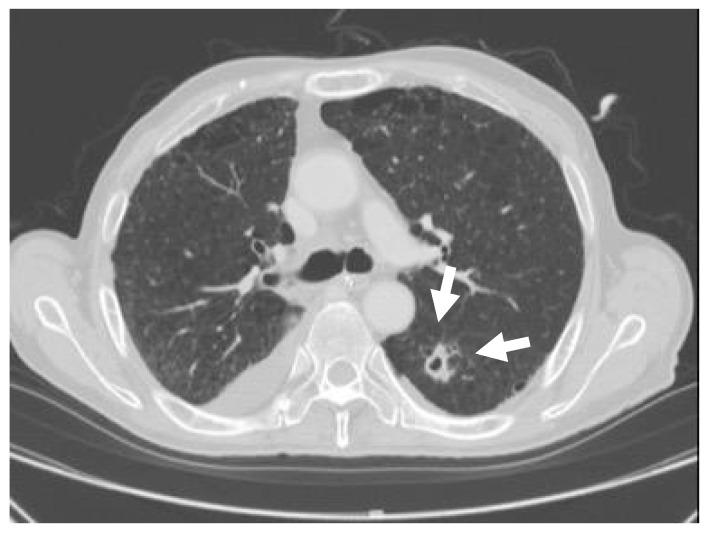
Cavitary nodule. The CT depicted a 20 mm cavitary nodule in the left lung S6.

**Figure 4 dentistry-07-00102-f004:**
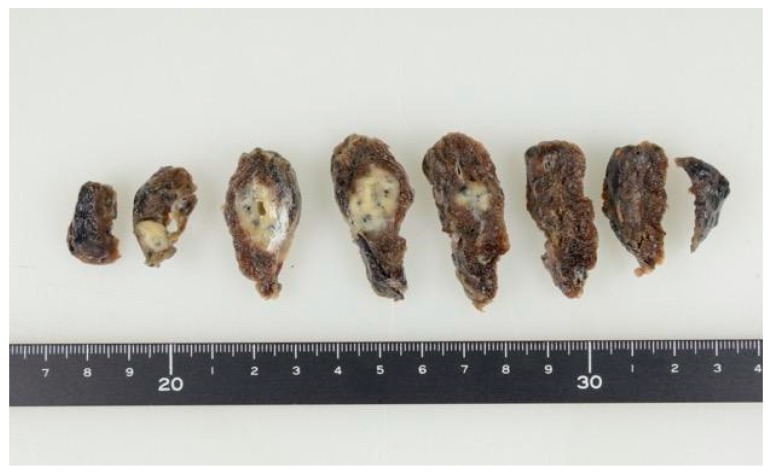
Resected specimen. A white solid mass with a charcoal powder deposition, measuring 20 × 15 × 11 mm, was resected.

**Figure 5 dentistry-07-00102-f005:**
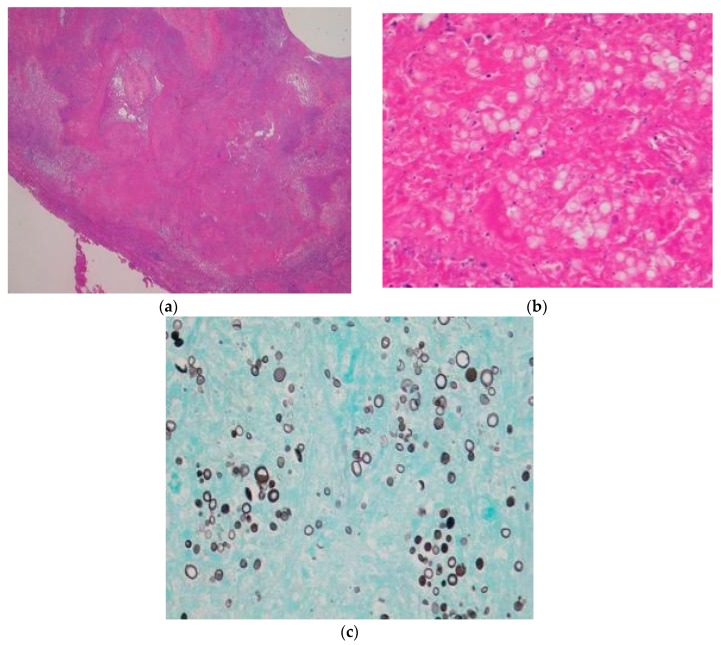
Pseudo-epithelial granuloma. A pseudo-epithelial granuloma with necrosis and many yeast-shaped fungi with capsules was identified. The encapsulated forms of Cryptococcus were revealed by Groccot staining. (**a**) HE staining (×12.5); (**b**) HE staining (×400); (**c**) Groccot staining (×400).

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
