# Peer review of "Pulmonary Nodule in a Patient with Oral and Lung Cancer: Cryptococcus Infection"

_dentistry, 2019, doi:10.3390/dj7040102_

Round 1

Reviewer 1 Report

The article describes the clinical case of pulmonary cryptococcosis after treatment of lung cancer and head and neck cancer. Cases of pulmonary cryptococcosis after or on the background of cancer are not unique, because the disease occurs against the background of inhibition of the immune system. moreover, the presence of one or two oncological diseases in the history does not affect the complexity of the diagnosis of cryptococcosis.
Remarks:
1) In the introduction, it would be nice to give statistics on the frequency of metastases in the lungs with head and neck tumors, as well as after radical treatment of lung cancer. How justified is the alertness of the doctor in this case?
2) The problem is not formulated in the article. Differential diagnosis is usually carried out between various lung diseases: non-specific inflammatory lung diseases (empyema, abscess, pneumonia), tuberculosis, sarcoidosis, benign tumors, lung cancer, lymphogranulomatosis, metastatic changes, inflammatory changes against the background of the tumor process in the lung tissue. The literature describes cases when a patient was diagnosed with large cell lung cancer, but could not correctly diagnose cryptococcosis. In my opinion, the focus in the article should not be on the description of such a case, but on the fact that mycological examination of the lung tissue should be included in the differential diagnosis algorithm used in the practice of a pulmonologist, thoracic surgeon, oncologist, and will allow timely detection of cryptococcosis of the lungs and choose the optimal tactics his treatment. Doctors can suspect cryptococcosis in a patient only in case of HIV infection, in all remaining cases this diagnosis is made by exclusion.

Therefore, I recommend redoing the introduction and revising the conclusions.

Author Response

Response to Reviewer 1 Comments

Point 1: In the introduction, it would be nice to give statistics on the frequency of metastases in the lungs with head and neck tumors, as well as after radical treatment of lung cancer. How justified is the alertness of the doctor in this case?

Response 1:
P1, L31-37 Added the frequency of metastasis from oral cavity and lung.

Point 2:The problem is not formulated in the article. Differential diagnosis is usually carried out between various lung diseases: non-specific inflammatory lung diseases (empyema, abscess, pneumonia), tuberculosis, sarcoidosis, benign tumors, lung cancer, lymphogranulomatosis, metastatic changes, inflammatory changes against the background of the tumor process in the lung tissue. The literature describes cases when a patient was diagnosed with large cell lung cancer, but could not correctly diagnose cryptococcosis. In my opinion, the focus in the article should not be on the description of such a case, but on the fact that mycological examination of the lung tissue should be included in the differential diagnosis algorithm used in the practice of a pulmonologist, thoracic surgeon, oncologist, and will allow timely detection of cryptococcosis of the lungs and choose the optimal tactics his treatment. Doctors can suspect cryptococcosis in a patient only in case of HIV infection, in all remaining cases this diagnosis is made by exclusion.

Response 2:
Thank you for your precious comments.
We agreed with your suggestions and revised the manuscript in coclusion.
P5, L147-150 Changed coclusion.

Reviewer 2 Report

This is an interesting case describing a pulmonary nodule in a patient with oral and lung cancer: cryptococcus infection.

It is a rare event and the paper is well documented with clinical, radiological and histological figure of good quality.

The authors consider the two localization (oral and lung) being primary tumors. One point that in my opinion should be discussed  is whether the lung lesion represents a metastatic carcinoma of the oral cancer (or why not).  

Author Response

Response to Reviewer 2 Comments

Thank you for your precious comments.

We agreed with your suggestions and revised the manuscript in coclusion.

Point 1: The authors consider the two localization (oral and lung) being primary tumors. One point that in my opinion should be discussed  is whether the lung lesion represents a metastatic carcinoma of the oral cancer (or why not).

Response 1:

P5, L137-138  Added the discussion.

Reviewer 3 Report

This manuscript reports about the clinical case showing a 68-year old Japanese male with lower gingival cancer and lung cancer, whose lung has a 20-mm cavity nodule after surgical treatments and was finally pathologically diagnosed as Cryptococcus infection. The authors present the interesting case, however, this report lacks some information for readers. Therefore, as described below, this manuscript needs to be addressed to some minor points to make the readers easily understand.

Line 53: Please spell out “SUV”. (Standard Uptake Volume)

Figures 2 and 3: Please put the arrows or arrowheads to indicate the pulmonary nodule on all panels.

Figure 5: Please describe the magnification of all images taken and put the bars showing the actual length on all panels as reference.

Author Response

Response to Reviewer 3 Comments

Thank you for your precious comments.

We agreed with your suggestions and revised the manuscript in coclusion.

Point 1: Line 53: Please spell out “SUV”. (Standard Uptake Volume)

Response 1:

P2, L66 Added “Standard Uptake Volume”.

Point 2:Figures 2 and 3: Please put the arrows or arrowheads to indicate the pulmonary nodule on all panels.

Response 2:

Figure 2 and 3 Putted arrows in figures.

Point 3:Figure 5: Please describe the magnification of all images taken and put the bars showing the actual length on all panels as reference.

Response 2:

Figure 5  Added the magnification for all pictures. I’m sorry we could not put bars of actual length.

Round 2

Reviewer 1 Report

In my opinion, the discussion should be more detailed. However, the previously made comments are corrected, so you can recommend accepting the article for publication.